

# Mediterranean Precipitation Response to Greenhouse Gases and Aerosols

Tao Tang[1], Drew Shindell[1], Bjørn H. Samset[2], Oliviér Boucher[3], Piers M. Forster[4], Øivind Hodnebrog[2], Gunnar Myhre[2], Jana Sillmann[2], Apostolos Voulgarakis[5], Timothy Andrews[6], Gregory Faluvegi[7], Dagmar Fläschner[8], Trond Iversen[9], Matthew Kasoar[5], Viatcheslav Kharin[10], Alf Kirkevåg[9], Jean-Francois Lamarque[11], Dirk Olivié[9], Thomas Richardson[4], Camilla W. Stjern[2], Toshihiko Takemura[12]

[1]Nicholas School of the Environment, Duke University, Durham, USA
[2]CICERO Center for International Climate and Environmental Research – Oslo, Norway
[3]Institute Pierre-Simon Laplace, Université Pierre et Marie Curie / CNRS, Paris, France
[4]University of Leeds, Leeds, United Kingdom
[5]Imperial College London, London, United Kingdom
[6]Met Office Hadley Centre, Exeter, UK
[7]Columbia University, New York, USA and NASA Goddard Institute for Space Studies, New York, USA
[8]Max-Planck-Institut für Meteorologie, Hamburg, Germany
[9]Norwegian Meteorological Institute, Oslo, Norway
[10]Canadian Centre for Climate Modelling and Analysis, Victoria, BC, Canada
[11]National Center for Atmospheric Research, Boulder, USA
[12]Kyushu University, Fukuoka, Japan

*Correspondence to*: Tao Tang (tao.tang@duke.edu)

**Abstract.** Atmospheric aerosols and greenhouse gases affect cloud properties, radiative balance and thus, the hydrological cycle. Observations show that precipitation has decreased in the Mediterranean since the 20th century, and many studies have investigated possible mechanisms. So far, however, the effects of aerosol forcing on Mediterranean precipitation remain largely unknown. Here we compare Mediterranean precipitation responses to individual forcing agents in a set of state-of-the-art global climate models (GCMs). Our analyses show that both greenhouse gases and aerosols can cause drying in the Mediterranean, and that precipitation is more sensitive to black carbon (BC) forcing than to well-mixed greenhouse gases (WMGHGs) or sulfate aerosol. In addition to local heating, BC appears to reduce precipitation by causing an enhanced positive North Atlantic Oscillation (NAO)/Arctic Oscillation (AO)-like sea level pressure (SLP) pattern, characterized by higher SLP at mid-latitudes and lower SLP at high-latitudes. WMGHGs cause a similar SLP change, and both are associated with a northward diversion of the jet stream and storm tracks, reducing precipitation in the Mediterranean while increasing precipitation in Northern Europe. Though the applied forcings were much larger, if forcings are scaled to those of the historical period of 1901-2010, roughly one-third ($31\pm17\%$) of the precipitation decrease would be attributable to global BC forcing with the remainder largely attributable to WMGHGs whereas global scattering sulfate aerosols have negligible impacts. The results from this study suggest that future BC emissions may significantly affect regional water resources, agricultural practices, ecosystems, and the economy in the Mediterranean region.



# 1 Introduction

Aerosols, fine particles in the atmosphere produced by both natural processes and anthropogenic activities, impact the Earth's climate by scattering and absorbing solar radiation (direct effect), or by modifying the properties of clouds (indirect effects) through a variety of mechanisms including atmospheric heating and changes in ice nuclei and cloud

condensation nuclei (CCN), including their size, location and concentration. These changes may significantly affect solar radiation and precipitation (*Ramanathan et al.*, 2001; *Kaufman et al.*, 2002; *Shindell et al.*, 2012; *Bond et al.*, 2013; *Boucher et al.*, 2013). The effects of aerosols on climate have been widely studied both on global and regional scales (*Ramanathan and Carmichael*, 2008; *Shindell and Faluvegi*, 2009). For example, *Menon et al.* (2002) reported slight cooling and drying trends in the northern part of China in the 2nd half of the 20th century, and attributed such

trends to the emissions of BC aerosols based on model simulations. Similarly, *Hodnebrog et al.* (2016) reported a precipitation decrease in southern Africa due to local biomass burning aerosols based on analyses of model simulations and local energy budget. On the other hand, *Koren et al.* (2012) argued that aerosols could intensify rainfall events in the lower and mid-latitudes by analyzing satellite observations. However, *Stevens and Feingold* (2009) contended that the effects of aerosols on clouds and precipitation are very limited due to the buffering effects of the climate system

itself. In addition to their influence on temperature and precipitation, aerosols may also affect large-scale atmospheric circulation. *Takahashi and Watanabe* (2016), based on a series of model simulations, suggested that the Pacific trade winds were accelerated partially by sulfate aerosols during the past two decades. *Dunstone et al.* (2013) reported that aerosols could modulate Atlantic tropical storm frequency due to aerosol-induced shifts in the Hadley circulation. These differing results suggest that aerosol effects on regional climate may depend on the aerosol types, seasons, and

regions of interest.

A decreasing precipitation trend in the Mediterranean area since the 20[th] century has been reported and its possible causes have been investigated in many studies (*Piervitali et al.*, 1998; *Buffoni et al.*, 1999; *Mariotti et al.*, 2002; *Dünkeloh and Jacobeit*, 2003; *Xoplaki et al.*, 2004). For instance, *Quadrelli et al.* (2001) observed a strong correlation

between winter Mediterranean precipitation and the NAO (*Hurrell et al.*, 2001). *Krichak and Alpert* (2005) suggested that the East Atlantic-West Russia (EA-WR) pattern may also play an important role in modulating the precipitation in the Mediterranean. Hence, the responses of Mediterranean precipitation to these large-scale variability patterns (e.g., NAO, EA-WR), and to some extent how these patterns might be responding to external drivers, are fairly well-understood (*Black et al.*, 2010). However, prior studies included all the drivers at once, so cannot discern the relative

roles of WMGHGs and other agents. Anthropogenic aerosols have been reported to greatly influence the temperature in the Mediterranean (*Nabat et al.*, 2014), but the effects of aerosols on Mediterranean precipitation have not been carefully examined. Since precipitation impacts water availability for both ecosystems and human societies, it is crucial to understand the different impacts of the climate drivers that are responsible for the Mediterranean precipitation trend. To bridge this knowledge gap, here we analyze Mediterranean precipitation changes based on a

group of state-of-the-art GCMs that examined the precipitation response to individual climate drivers, which could help inform management of water resources, regional societal activities such as agriculture, and even emissions mitigation.





## 2 Data and Method

### 2.1 Data

This study employed output from nine models participating in the Precipitation Driver and Response Model Intercomparison Project (PDRMIP), utilizing simulations examining the individual responses to $CO_2$, sulfate and BC

aerosols. In these simulations, perturbations were performed with each model at global scale: a doubling of $CO_2$ concentration ($CO_2 \times 2$), 10 times present-day BC concentration ($BC \times 10$), and 5 times present-day $SO_4$ concentration ($SO_4 \times 5$). All perturbations were abrupt. $CO_2 \times 2$ perturbations were applied relative to the models' own baseline values. For aerosol perturbations, monthly present-day concentrations were derived from the AeroCom Phase II initiative (*Myhre et al.*, 2013a). The concentrations were multiplied by the stated factors. A few models instead perturbed aerosol

emissions, in most cases again using AeroCom Phase II data. Each perturbation was run in two configurations, a 15-yr fixed sea surface temperature (SST) simulation and a 100-yr coupled simulation. Each fixed-SST simulation is compared with its fixed-SST control simulation to diagnose the effective radiative forcing (ERF) due to each perturbation (*Myhre et al.*, 2013b), whereas each coupled run is compared with its coupled control run to examine climate response. The nine models used in this study are listed in Table 1. A more detailed description of PDRMIP

and its initial findings are given in *Samset et al.* (2016) and *Myhre et al.* (2017).

### 2.2 Method

In addition to direct analysis of meteorological fields (e.g. precipitation, sea-level pressure) in the models, we also analyse the energy budget associated with the hydrological cycle. Following *Hodnebrog et al.* (2016) and *Muller and O'Gorman* (2011), the precipitation change is related to diabatic cooling and the horizontal transport of dry static

energy as follows:

$$L_c \Delta P = \Delta Q + \Delta H \tag{1}$$

Here $L_c$ is the latent heat of condensation of water vapor, which is 29 W m$^{-2}$ mm$^{-1}$ day. $\Delta P$ is the precipitation change.

$\Delta H$ is the column-integrated dry static energy flux divergence and $\Delta Q$ is the column-integrated diabatic cooling, which is calculated as:

$$\Delta Q = \Delta LW + \Delta SW - \Delta SH \tag{2}$$

where $\Delta LW$ is the change of longwave radiation in the atmospheric column and $\Delta SW$ is the change of shortwave radiation in the atmospheric column. $\Delta SH$ is the change of upward sensible heat flux.

Since most of the precipitation events occur in the wet season (Oct-Mar) in the Mediterranean, roughly 70% of total annual precipitation (*Mariotti et al.*, 2002; *Kottek et al.*, 2006), the analysis was restricted to the wet season in the

current study unless noted otherwise. All of the data used in this study were re-gridded into 2.5° × 2.5° horizontal resolution for analyses.





## 3 Results

Fig. 1 shows the multi-model mean (MMM) of normalized $\Delta P$ for each forcing. Both $CO_2$ and BC caused a substantial drying over Mediterranean, with a larger magnitude from BC (Fig. 1a-b) whereas $SO_4$ contributed very little in the Mediterranean region (Fig. 1c). Moreover, in stark contrast to the drying of the Mediterranean, Northern Europe shows

increasing precipitation trends for $CO_2$ and BC (Fig. 1a-b), which will be discussed in more details later. To compare the precipitation response quantitatively, the domain-averaged (purple rectangle in Fig. 1, 30°N-45°N, 10°W-40°E) trends are shown in Fig. 2. For $CO_2$, all of the nine models show drying trends (Fig. 2a). The MMM is -0.03±0.03 mm/day per $W/m^2$, with individual model values ranging from -0.01 to -0.06 mm/day per $W/m^2$. For BC (Fig. 2b), all nine models again show drying trends, with the MMM value -0.12±0.07 mm/day per $W/m^2$, which is four times as

large as that of $CO_2$. When it comes to $SO_4$ (Fig. 2c), the model results even differ in the sign of change and the MMM value is small (-0.01±0.04 mm/day per $W/m^2$). These analyses show that the precipitation response is more sensitive to BC forcing than to $CO_2$ and $SO_4$ in this region.

In order to investigate the mechanisms governing the precipitation response, we performed an energy budget analysis

for this region (Fig. 3). For $CO_2$, the drying is dominated by horizontal energy transport (gray box in the $CO_2$ panel), albeit some offset by diabatic cooling (pink box in the $CO_2$ panel). For BC, the net radiation change, which is primarily SW (red box in the BC panel), has a larger impact than the horizontal energy change (gray box in the BC panel), but the latter is nonetheless a substantial fraction of the net change. When it comes to $SO_4$, the small precipitation response results from the offsetting of net radiation change (pink box in the $SO_4$ panel) and horizontal energy transport (gray

box in the $SO_4$ panel). The energy budget analysis implies that the dynamical responses to $CO_2$ and BC played a crucial role in modulating the precipitation in this region.

We then analyzed the $\Delta SLP$ from the model output (Fig. 4). Specifically, it is seen that $CO_2$ induced strong SLP changes. The SLP increased at mid-latitudes, with increases extending from the North Atlantic to Southern Europe,

and decreased at high latitudes (Fig. 4a). BC led to a similar pattern of SLP change, but with increased magnitude (Fig. 4b), characterized by two increases centered in Europe and the Western North Atlantic. Compared with $CO_2$ and BC, $SO_4$ caused an opposite change (Fig. 4c). The $CO_2$ and BC forcings appear to induce a pattern similar to the positive phase of the NAO /AO (*Lorenz*, 1951), in which the jet streams and storm tracks are displaced northward, leading to a drier Mediterranean and precipitation increases in Northern Europe (Fig. 1a-b). Such a shift in response

to forcings is more clearly seen in the changes of zonal winds (Fig. 5). The $CO_2$ caused a strengthening of zonal winds in the whole upper atmosphere and a strengthening around 60°N from the near-surface to the top of the atmosphere, as well as weakening around 30°N from the near-surface to the mid-troposphere (Fig. 5a), as in prior studies (*Shindell et al.*, 2001) The strengthening around 60°N is more apparent for BC (Fig. 5b). Similar results were seen in response to aerosol forcing in a prior study (*Allen and Sherwood*, 2011). This shift is possibly due to the enhancement of the

tropospheric temperature gradient between mid-latitudes and high-latitudes, as suggested by *Allen et al.* (2012).





Our analyses illustrate that BC aerosols may modulate regional precipitation in part via modifying large-scale circulation patterns. Many previous studies suggest that BC could impact regional precipitation by changing the local vertical temperature profile, in which BC aerosols absorb solar radiation and heat the atmosphere, thus suppressing convection and cloud formation (*Kaufman et al.*, 2002; *Meehl et al.*, 2008; *Ramanathan and Carmichael*, 2008;

*Hodnebrog et al.*, 2016). Our results (analyses of the energy budget, SLP and zonal winds) suggest that a portion of the drying is also associated with large-scale circulation responses. In addition, our pattern of jet stream/storm track changes (Fig. 4 & 5) is also in agreement with the projections from the latest IPCC report (*Collins et al.*, 2013) based on a set of CMIP5 models, with increasing storm activities in Northern Europe and decreasing storms in the Mediterranean. Such a shift of storm tracks may further reduce the precipitation in the Mediterranean, though

reductions in WMGHG or BC emissions may help to mitigate the projected drying.

**4 Case Study – Historical Observations and Scaled Model Results**

The above analyses demonstrated how the precipitation and circulation responded to each forcing both qualitatively and quantitatively. In order to explore their potential relative contributions to the total precipitation change, here we apply linear scaling to the model output. Since PDRMIP utilized large aerosol and greenhouse gas changes in order

to achieve strong signals that could be statistically significant with a relatively modest amount of computational time, the precipitation change from those model outputs needs to be scaled in order to compare with observations. Uncertainties related to this approach are discussed further in section 5.

In this study, we focus on the period from 1901 to 2010. The scaled precipitation change for each individual forcing

is defined as:

$$\Delta P_{scaled} = \Delta P \times (ERF_{1901\text{-}2010} \, / \, ERF_{model}) \tag{3}$$

In equation (3), $\Delta P$ is the precipitation change over Mediterranean in the model during the last 50 years in the coupled

run, since the model has reached near-equilibrium state after 30 years. $ERF_{1901\text{-}2010}$ is the historical global ERF for the period of 1901-2010. The values were obtained from the latest Intergovernmental Panel on Climate Change (IPCC) assessment report (*Myhre et al.*, 2013b). The $ERF_{1901\text{-}2010}$ value used for $CO_2$ is 2.33 W/m$^2$, which is larger than the $CO_2$ value from the IPCC report as $CO_2$ was used to represent all WMGHGs in this case study. $ERF_{1901\text{-}2010}$ values for BC and $SO_4$ are 0.28 W/m$^2$ and -0.33 W/m$^2$, respectively. $ERF_{model}$ is the global ERF in the PDRMIP models, which

was obtained by calculating the energy flux change at the top of the atmosphere from years 6 to 15 of the fixed-SST simulations, since present models largely equilibrate within 5 years of fixed-SST running (*Kvalevåg et al.*, 2013). In addition to the direct effects of the aerosols, the indirect effects of aerosols were also included in most of the models and thus, in the $ERF_{model}$. The value of ($ERF_{1901\text{-}2010}$ / $ERF_{model}$) is the scaling factor applied to model precipitation output to match historical forcing levels. They are 0.64 [0.57, 0.69], 0.33 [0.10, 0.68] and 0.11 [0.04, 0.16] for $CO_2$,

BC and $SO_4$, respectively (where the values indicate the mean [min, max] across the nine models). An important assumption here is that the $\Delta P$ changes linearly with $ERF_{model}$.




$\Delta P_{scaled}$ is calculated for $CO_2$, BC, and $SO_4$ separately. The total $\Delta P_{scaled}$ is the combination of the three, assuming their responses to those forcings can be added linearly. It should be noted that in this analysis, we use the combined responses to WMGHGs, BC and $SO_4$ to approximate the total historical response over 1901-2010. Several additional

factors may have also played a role, including natural forcing (solar and volcanic activities), land use/land cover change, contrails, ozone ($O_3$) (both tropospheric and stratospheric) and stratospheric water vapor, which have forcings of -0.03, -0.09, 0.05, 0.26 and 0.06 W/m$^2$, respectively (*Myhre et al.*, 2013b). As all these forcings are fairly small, simulations to isolate their impacts would be extremely computationally expensive and hence were not performed but to first order we expect their exclusion is unlikely to greatly affect our results. Characterization of the influence of

these other drivers merits future study, particularly as some operate via different physical processes (e.g. tropospheric ozone is both a greenhouse gas and an absorber of incoming solar radiation). Similar analyses were also performed to obtain scaled SLP change ($\Delta SLP_{scaled}$), zonal wind change and energy budget change in the atmospheric column.

Several observational and reanalysis datasets were also employed in this part of our study. For precipitation, Global

Precipitation Climatology Center (GPCC) monthly precipitation data (*Schneider et al.*, 2011), provided by NOAA/OAR/ESRL from their website (https://www.esrl.noaa.gov/psd/data/gridded/data.gpcc.html#detail), is employed. It is a high quality gridded dataset that is mainly terrestrial station-derived. For SLP, we use HadSLP2 data (*Allan and Ansell*, 2006), which is created by combining marine observations from ICOADS data (*Worley et al.*, 2005) and land (terrestrial and island) observations (available at

https://www.esrl.noaa.gov/psd/data/gridded/data.hadslp2.html). We also use NCEP/NCAR reanalysis data (*Kalnay et al.*, 1996), downloaded from https://www.esrl.noaa.gov/psd/data/gridded/data.ncep.reanalysis.derived.surface.html, for the comparisons of zonal wind. All these datasets have undergone rigorous quality control and have been widely used in the climate community, including the IPCC 2013 assessment report (*Hartmann et al.*, 2013). The trends of the observed and reanalysis data were estimated by a simple linear regression applied to the same period of the datasets.


The combination of WMGHGs, BC and $SO_4$ exerted a strong drying trend in the Mediterranean (Fig. 6a). The drying trends shown here are statistically significant and consistent with the observations (Fig. 6b), as well as previous studies (*Buffoni et al.*, 1999; *Mariotti et al.*, 2002; *Dünkeloh and Jacobeit*, 2003). When averaged over the whole domain, the scaled drying trends caused by WMGHGs, BC and $SO_4$ are -1.28±1.21 mm/decade, -0.58±0.34 mm/decade and -

0.03±0.21 mm/decade, respectively (not shown here). When combined (Fig. 6c), all nine models show decreased precipitation, with MMM value of -1.89±1.39 mm/decade, which is roughly a 5% decrease relative to the climatology of the control simulations. Such a decreasing trend is indistinguishable from the observations (-2.78±1.10 mm/decade, a 10% decrease compared with its 110-yr climatology). In spite of the dominant role of WMGHGs in the drying of the Mediterranean, BC contributed roughly one-third (31±17%) of the total forced precipitation decrease in this region

whereas the contribution of the scattering aerosol-$SO_4$ is negligible (~1.6%). We also examined the trend of precipitation in the control simulations and found only very weak responses (Fig. 6c), with a mean value of 0.004±0.03 mm/decade and maximum value of 0.03 mm/decade in any individual model. Since current GCMs are able to capture



the broad spatial and temporal features of internal variability (*Flato et al.*, 2013), and the forced drying signal is almost equal to the total signal (Fig. 6a-c), the consistent drying trend in the models is very unlikely to be attributable to unforced variability and appears realistic. The energy budget change (Fig. 6d) clearly shows that the net precipitation decrease is mainly due to horizontal energy transport (gray box) rather than diabatic cooling (pink box), because the
absorption of SW radiation (red box) and LW radiative cooling (green box) offset one another in total.

Fig. 7a shows the overall response of SLP to these forcings, with strong SLP increases at mid-latitudes and strong decreases at higher latitudes. Such patterns of SLP changes are also found in the observed datasets (Fig. 7b), albeit with a larger magnitude. The combined pattern of zonal wind responses shows winds intensified at the northern edge
of the jet stream and weakened at the southern edge (Fig. 7c). The NCEP dataset depicts a similar pattern of changes, with winds intensifying at 60°N and weakening at 30°N, but as with SLP, with a stronger magnitude (Fig. 7d). Some previous studies have pointed out that current GCMs significantly underestimate the tropical expansion and jet stream shift, which could be related to the short observational record, large internal variability or model deficiencies (*Johanson and Fu*, 2009; *Allen et al.*, 2012). Despite the underestimations, our analyses clearly demonstrate the shift
of the jet stream in response to these forcings that appears qualitatively consistent with observations.

Based on the model simulations in the current study, the pattern of climate response to BC forcing over the past ~110 years is similar to the response to WMGHGs over Europe and the North Atlantic, including precipitation, SLP and zonal winds. At the same time, our results suggest that $SO_4$ played a very limited role in modulating Mediterranean
precipitation trends and North Atlantic storm tracks. In other words, the precipitation trends during the past 110 years in the Mediterranean are likely to be only weakly sensitive to scattering aerosols that were not modeled (e.g., organic carbon, nitrate) or the uncertainties in aerosol negative forcing (probably not even for indirect forcing, as they were included in sulfate simulations for most models). The small sensitivity of $SO_4$ is likely due to compensation between local and remote effects (*Liu et al.*, 2018). Combined with its small *ERF*, the role of $SO_4$ appears to be negligible
during this period. However, the simulations examined here were not designed to determine whether the aerosol effects are due to local or remote emissions from the models. Initial analysis from PDRMIP regional experiments (in which BC over Asia only is multiplied by 10, with everything else being held at present-day levels) indicate that BC from Asia contributes as much as 60% to the drying signal in the Mediterranean, and in fact a larger average rainfall change in the Mediterranean than averaged over Asia itself. This suggests that the remote effects of BC may have dominated
the Mediterranean precipitation changes. Hence the response to global BC increases may be a reasonable proxy for the 20[th] century changes, although it would be useful to explore the effects of local reductions from Europe itself in the late 20[th] century. The relative impacts of local versus remote forcing will be further explored in forthcoming PDRMIP analyses.

## 5 Discussion and Conclusion

Since PDRMIP experiments are equilibrium simulations while the real-world is transient, and we scaled PDRMIP forcing to match historical levels, we examined related experiments to test both these aspects of the methodology used



in our comparison with historical observations. Historical GHG-only simulations using the same CMIP5 models (*Taylor et al.*, 2012) that participated in the PDRMIP project were collected and analyzed (data available at http://strega.ldeo.columbia.edu:81/CMIP5/.monthly/.byModel/). Six models are available and each model has 1-5 ensemble members. All of the six models show drying trends (Fig. 8), with a MMM value of -1.32±1.65 mm/decade

(-1.29 mm/decade when weighted by ensemble size) which is quite close to the WMGHGs result of our scaled equilibrium PDRMIP output (-1.28±1.21 mm/decade). In fact, the overlap of their probability density functions is 0.85, assuming a normal distribution. This comparison indicates that our methodology does not appear to be a large source of uncertainty in the current study, though response to other agents may not be as linear as those to WMGHGs (unfortunately, simulations are not currently available to evaluate other forcers, and, given the enormous expense of

running enough ensemble members to isolate the relatively small signals for individual aerosols, are unlike to be anytime soon). Similar analyses were also performed for SLP and zonal winds, and again there is no appreciable difference between the historical transients and the scaled equilibrium responses. The consistent results suggest that the methodology works surprisingly well.

In addition to the wet season, precipitation during the dry season (Apr-Sep) for the PDRMIP model was also analyzed. The modelled dry season precipitation trends, however, do not match the observations well (not shown). The modelled results also show a statistically strong drying trend while the observations do not show significant changes. Two possible reasons may be responsible for the apparent discrepancies. One is that only 30% of the total precipitation occurs during the dry season (boreal summer months) and it is difficult to simulate the uneven distribution of

infrequent rainfall events. The other is that there are large uncertainties in the observational data itself. Unlike the wet season, in which nearly half of the grid boxes show statistically significant trends (Fig. 6b), almost none of the grid boxes show statistically significant trends in the dry season, undermining the robustness of the observational results.

The drying influence of WMGHGs will be more prominent in the future due to their projected continued growth. In

contrast, many studies suggest that aerosol concentrations may decrease rapidly in the future due to air quality and climate policies along with their relatively short lifetime compared with WMGHGs (*Andreae et al.*, 2005; *Myhre et al.*, 2013b; *Shindell et al.*, 2013). Reductions of BC could, to some extent, slow down the drying trend in the Mediterranean. Overall, a drier Mediterranean region is expected owing to increasing WMGHGs, but the pace of change in global BC emissions may substantially modify the drying rate in the near term.


Some limitations and uncertainties still exist in our current study. First, it is important to keep in mind that the case study in Section 4 is not a formal attribution analysis, despite the estimation of BC contribution. Our aim is to give a first grasp of the effects of aerosol on regional precipitation in the Mediterranean. Second, although our comparison of scaled equilibrium and unscaled transient simulations indicates that our methodology works well at least for

WMGHGs, there is no systematic study so far exploring the linearity (or non-linearity) of the precipitation responses to BC or the linearity of responses to multiple versus individual forcings. Third is that the $ERF_{1901-2010}$ of BC represents direct effects only (*Myhre et al.*, 2013b). Semidirect and indirect effects, however, are included in many of our





PDRMIP models, and thus in $ERF_{model}$. We did not include these effects in the scaling in this study for two reasons: first, the indirect effects of BC in the PDRMIP models do not include ice particles and are difficult to evaluate as BC concentrations were prescribed so that they cannot interact fully with clouds, indicating that they are not fully resolved, and second, the net $ERF_{1901-2010}$ of semidirect plus indirect effects is likely small (-0.1 to +0.2 W/m$^2$) with a very large

overall uncertainty range (-0.4 to +0.9 W/m$^2$) (*Bond et al.*, 2013). If the semi-direct and indirect effects of BC (-0.1 to +0.2 W/m$^2$) are considered in the scaling, the $\Delta P_{scaled}$ of BC aerosol would be -0.44 to -0.87 mm/decade and still contribute a substantial part (25 to 40 %) to the drying. The situation is similar for sulfate aerosol, for which indirect effects are included in $ERF_{model}$, but not in $ERF_{1901-2010}$. We did not include indirect effects in our scaling as these were not attributed to individual aerosol species in the IPCC AR5 (*Boucher et al.*, 2013). If the indirect effects are

considered, the negative $ERF_{1901-2010}$ could increase roughly by a factor of two (assuming indirect effects are largely associated with sulfate). However, the $\Delta P_{scaled}$ of sulfate aerosol would still be very small compared with WMGHGs or BC, which would not impact our conclusions. The final issue is related to the design of the model simulations. The perturbations are 5× or 10× present-day aerosol concentrations, which are time-invariant. The aerosols, however, have significant spatial and temporal variations. For instance, aerosol concentrations have been increasing in Asia

continuously since 1950, but decreasing in Europe since the 1970s (*Allen et al.*, 2013). As noted, further work is needed to determine how much of the Mediterranean trends result from local relative to remote forcing. To the extent that the trends are driven by remote forcing the potential influence of such spatio-temporal variations will be small. This will be explored in future PDRMIP simulations.

Our analyses show that both WMGHGs and BC influence wet season Mediterranean rainfall by causing an enhanced positive NAO/AO-like SLP pattern as well as by some local heating due to SW absorption. The SLP pattern is characterized by higher SLP in the North Atlantic and Mediterranean and lower SLP in the Northern part of Europe, which diverts the jet stream and storm tracks further northward, reducing the precipitation in the Mediterranean and increasing precipitation in Northern Europe. In contrast, global perturbations of the scattering aerosol SO$_4$ have a

negligible impact. The results from this study may have important implications to the management of regional water resources, agricultural practice, ecosystems, environment, and economics as well as social development and behavior in a warming climate. They also stress the importance of accounting for the aerosols (and generally short-lived forcers) for short-term (e.g., decadal) regional climate prediction.

**Acknowledgement**

All model results used for this study are available to the public through the Norwegian NORSRORE data storage facility. We thank the three reviewers for their insightful comments. We also acknowledge the NASA High-End Computing Program through the NASA Center for Climate Simulation at Goddard Space Flight Center for computational resources to run the GISS-E2R model and support from NASA GISS. PDRMIP is partly funded through the Norwegian Research Council project NAPEX (project number 229778). O. Boucher acknowledges HPC resources

from CCRT under the gencmip6 allocation provided by GENCI (Grand Equipement National de Calcul Intensif). P. Forster and T. Richardson were supported by NERC grants NE/K007483/1 and NE/N006038/1. Ø. Hodnebrog was





partly funded through the Norwegian Research Council project HYPRE (project no. 243942). A.Voulgarakis and M.Kasoar are supported by NERC under grant NE/K500872/1. Simulations with HadGEM3-GA4 were performed using the MONSooN system supplied under the Joint Weather and Climate Research Programme of the Met Office and NERC. D. Olivié, A. Kirkevåg and T. Iversen were supported by the Norwegian Research Council through the projects EVA (grant 229771), EarthClim (207711/E10), NOTUR (nn2345k), and NorStore (ns2345k). T. Takemura was supported by the supercomputer system of the National Institute for Environmental Studies, Japan, the Environment Research and Technology Development Fund (S-12-3) of the Ministry of the Environment, Japan and JSPS KAKENHI Grant Numbers 15H01728 and 15K12190. Computing resources for CESM1-CAM5 (ark:/85065/d7wd3xhc) simulations were provided by the Climate Simulation Laboratory at NCAR Computational and Information System Laboratory, sponsored by the National Science Foundation and other agencies.

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





**Table 1. Descriptions of the nine PDRMIP models (adapted from *Liu et al.* (2018)). GA = Global Atmosphere. HTAP2 = Hemispheric Transport Air Pollution, phase 2.**

| Model | Version | Resolution | Ocean setup | Aerosol setup | References |
|---|---|---|---|---|---|
| Canadian Earth System Model (CanESM2) | 2010 | 2.8×2.8 35 levels | Coupled | Emission | *Arora et al.* (2011) |
| Goddard Institute for Space Studies Model E2, coupled with Russell ocean model (GISS-E2-R) | E2-R | 2×2.5 40 levels | Coupled | Fixed concentration | *Schmidt et al.* (2014) |
| Hadley Center Global Environment Model, version 2-Earth System (HadGEM2-ES) | 6.6.3 | 1.875×1.25 38 levels | Coupled | Emissions | *Collins et al.* (2011) |
| HadGEM3 | GA 4.0 | 1.875×1.25 85 levels | Coupled | Fixed concentration | *Bellouin et al.* (2011) *Walters et al.* (2014) |
| Model for Interdisciplinary Research on Climate-Spectral Radiation-Transport Model for Aerosol Species (MIROC-SPRINTARS) | 5.9.0 | T85 (approx. 1.4×1.4 ), 40 levels | Coupled | HTAP2 emissions | *Takemura et al.* (2005) *Takemura et al.* (2009) *Watanabe et al.* (2010) |
| Community Earth System Model, version 1 (Community Atmosphere Model, version 4) [CESM1-CAM4] | 1.0.3 | 2.5×1.9 26 levels | Slab | Fixed concentration | *Neale et al.* (2010) *Gent et al.* (2011) |
| CESM1-CAM5 | 1.1.2 | 2.5×1.9 30 levels | Coupled | Emissions | *Hurrell et al.* (2013) *Kay et al.* (2015) *Otto-Bliesner et al.* (2016) |
| Norwegian Earth System Model (NorESM) | NorESM1-M (intermediate resolution) | 2.5×1.9 26 levels | Coupled | Fixed concentration | *Bentsen et al.* (2013) *Iversen et al.* (2013) *Kirkevåg et al.* (2013) |
| L'institut Pierre-Simon Laplace Coupled Model, version 5A (IPSL-CM5A) | CMIP5 | 3.75×1.875 39 levels | Coupled | Fixed concentration | *Dufresne et al.* (2013) |





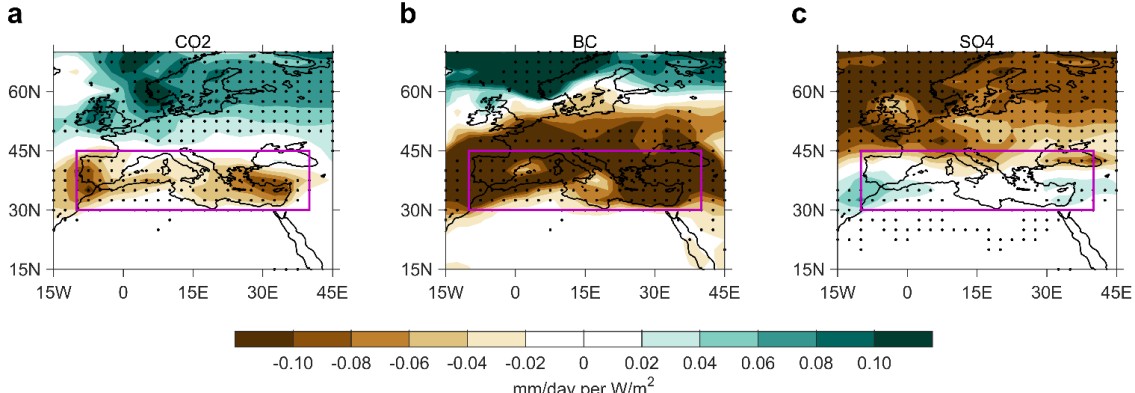

**Figure 1: Normalized $\Delta P$ (change per unit forcing) for (a) CO₂, (b) BC and (c) SO₄. Black dots indicate the change is significant at 0.95 confidence level. Please note that the sign for SO₄ is flipped due to its negative forcing. Thus, the results shown for SO₄ is the precipitation change per unit negative forcing.**





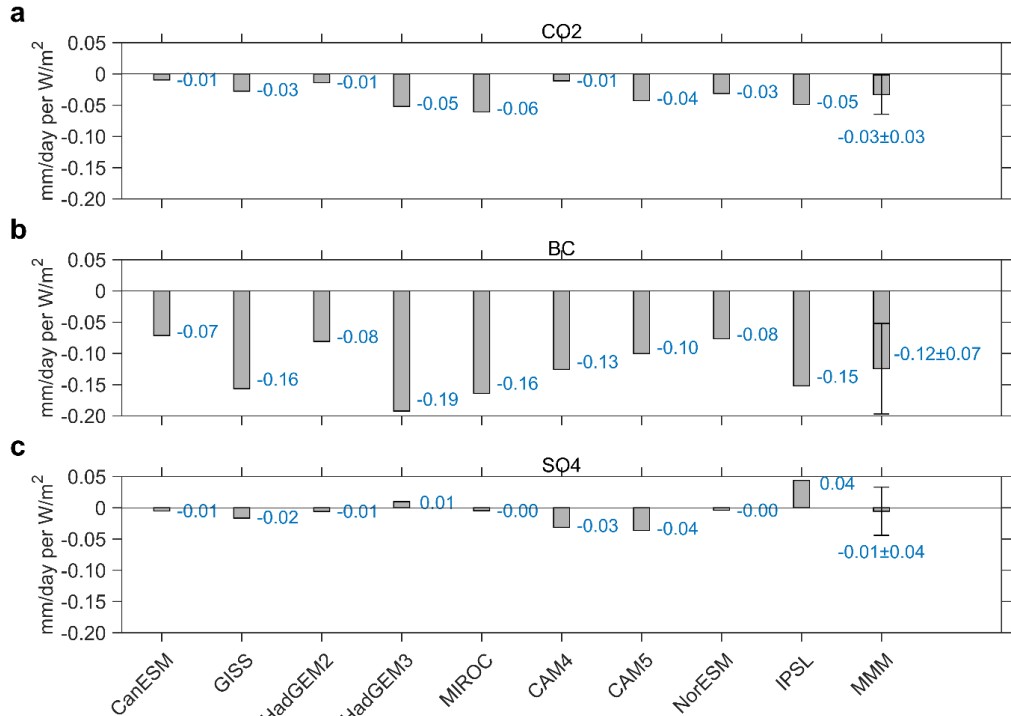

**Figure 2: Domain-averaged ΔP (purple rectangles in Fig. 1) for (a) CO2, (b) BC and (c) SO4. Error bars of multi-model mean (MMM) are 90% inter-model spread.**



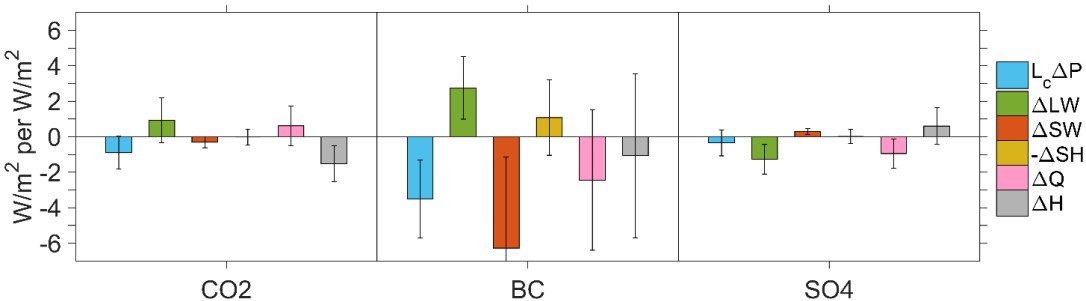

**Figure 3: Domain-averaged (purple rectangles in Fig. 1) energy budget change for each forcing and energy component as shown in Equation (1) and (2). It holds that $L_c\,\Delta P$ (blue) = $\Delta Q$ (pink) + $\Delta H$ (gray), where $L_c\,\Delta P$ is the change in total latent heating, $\Delta Q$ (pink) = $\Delta LW$ (green) + $\Delta SW$ (red) - $\Delta SH$ (brown) is the change in diabatic cooling of the atmospheric column due to shortwave and longwave radiation, and sensible heat flux, $\Delta H$ is the change in column-integrated dry static energy flux divergence. The error bars indicated 90% inter-model spread.**



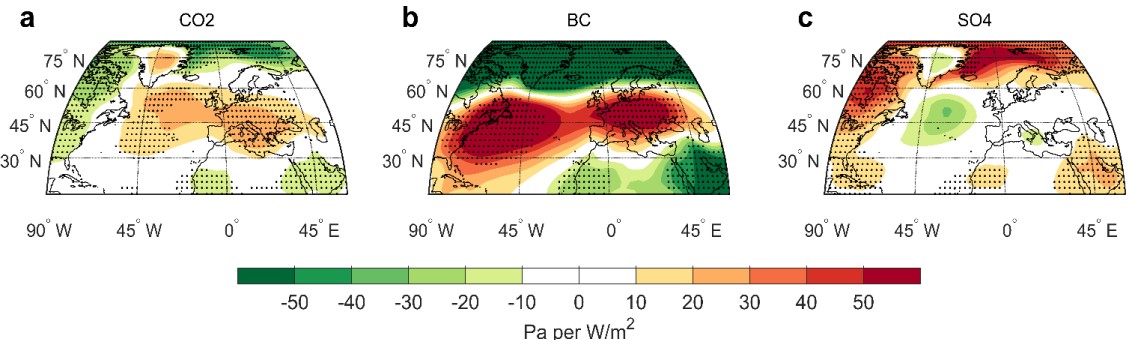

**Figure 4: Same as Fig. 1, but for sea level pressure (SLP).**





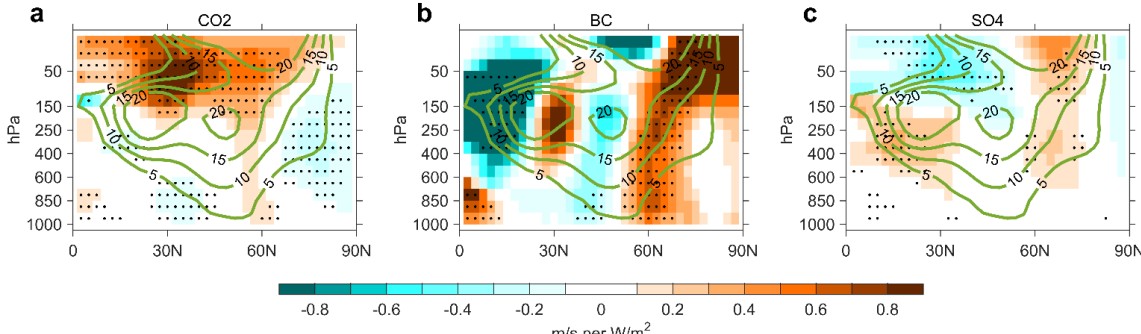

**Figure 5: Same as Fig. 1, but for zonal wind. The green contours represent the 50-yr climatology of the zonal wind in the control simulations. The contours are at the interval of 5m/s, with positive values indicating eastward winds.**



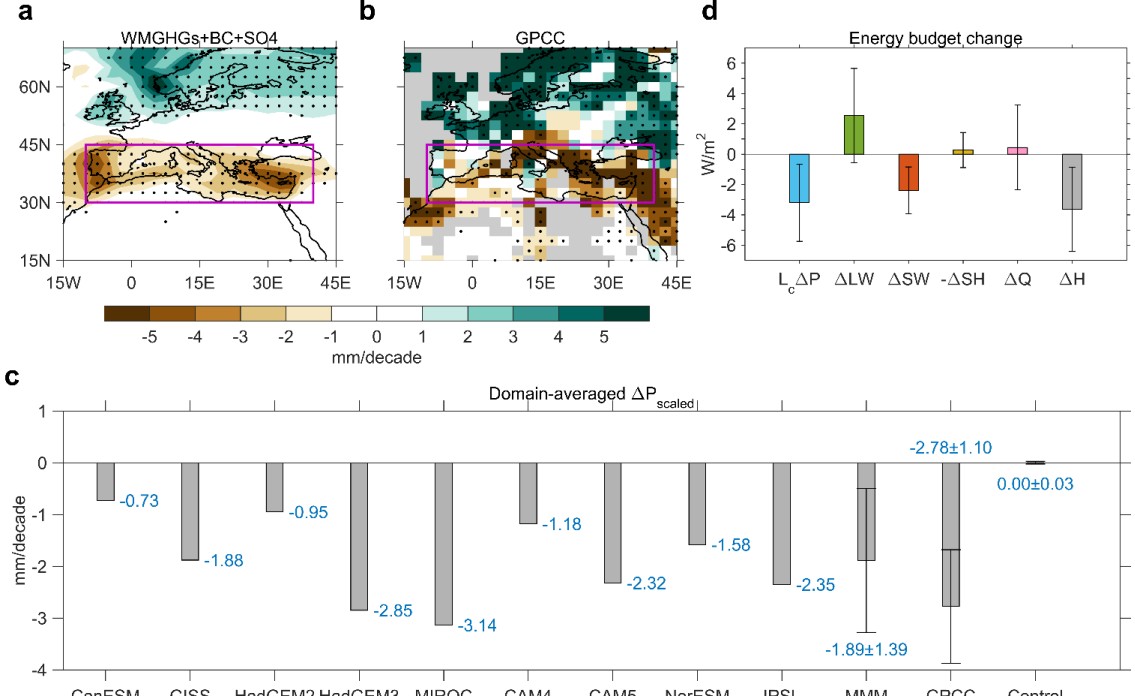

**Figure 6: Scaled change for the combination of WMGHGs, BC and SO₄ for (a) spatial pattern of precipitation, (c) domain-averaged precipitation change, and (d) energy budget change, along with (b) GPCC observational data (for which gray indicate missing or incomplete data). The dots in (a) and (b) indicate changes are significant at 0.95 and 0.9 confidence level, respectively. Error bars in (c) and (d) indicate 90% uncertainty ranges.**





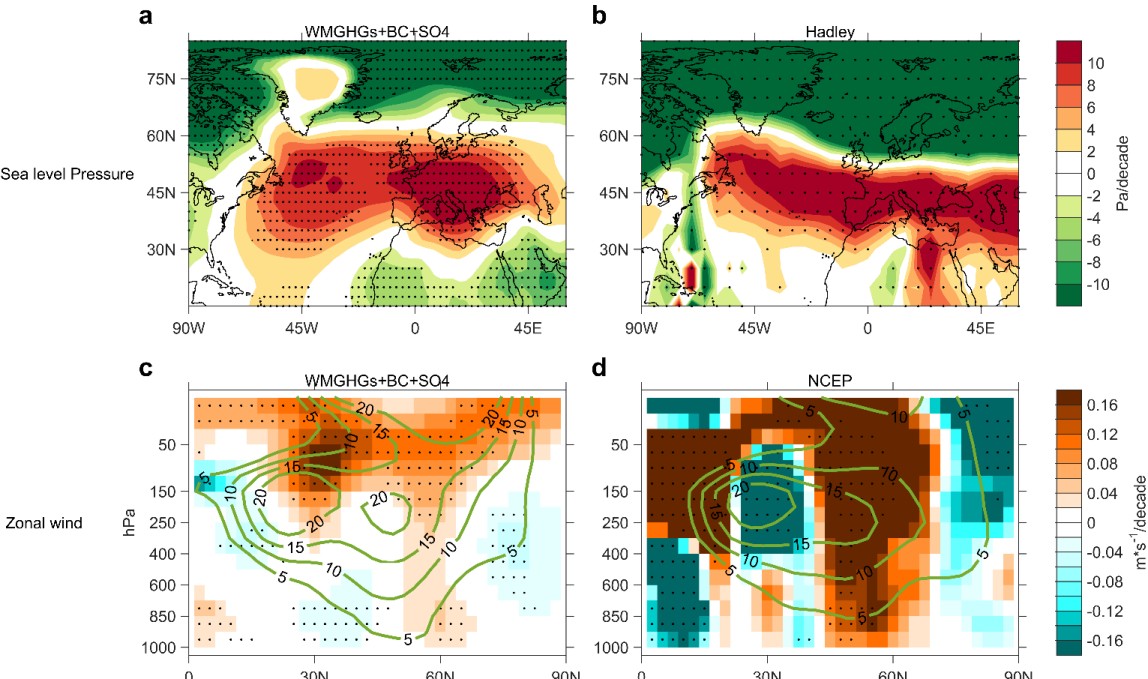

**Figure 7: SLP (a & b) and zonal wind change (c & d). (a) and (c) are scaled change for the combination of WMGHGs, BC and SO4, while (b) and (d) are Hadley observational data and NCEP reanalysis data, respectively. Dots indicate the changes are significant at 0.95 confidence level. The green contours in (c) and (d) represent the climatology position of the zonal wind. The contours are at the interval of 5m/s, with positive values indicating eastward winds.**





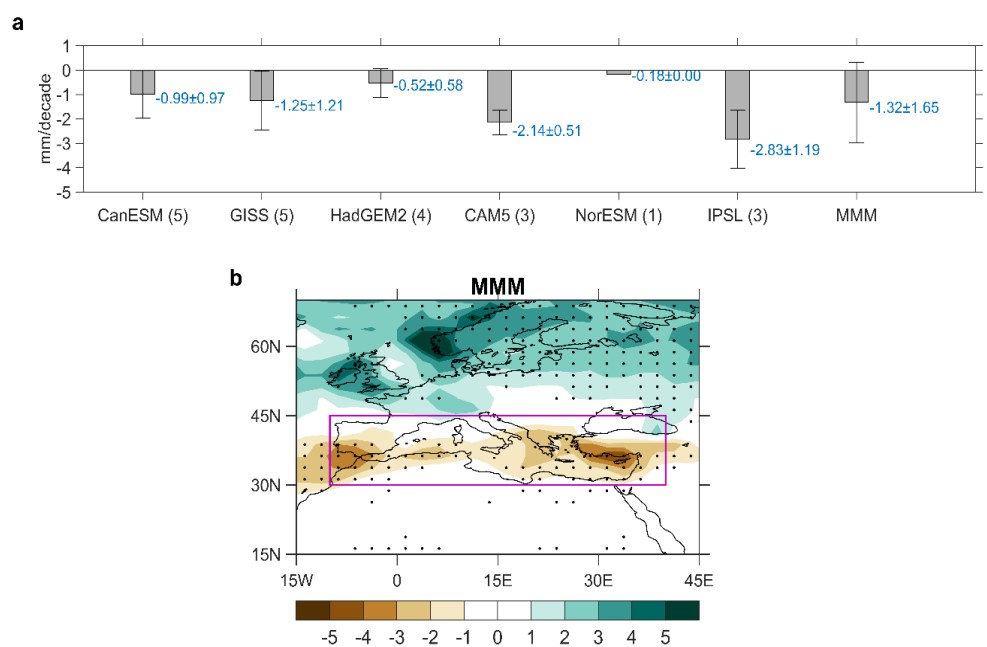

**Figure 8: Precipitation trends of CMIP5 HistoricalGHG simulations during 1901-2005, (a) domain-averaged (purple rectangles in Fig. 1) trends for each model. Error bars of each model indicate 90% inter-ensemble spread and error bars of MMM indicate the 90% inter-model spread. The numbers in parentheses indicate the ensembles collected for each model. (b) spatial pattern of MMM value for the trends. Dots indicate that the change is significant at 0.95 confidence level.**