# Peer review of "Dynamical Response of Mediterranean Precipitation to Greenhouse Gases and Aerosols"

_Atmospheric Chemistry and Physics, 2018_

## Referee Comment (RC1) · Anonymous Referee #1 · 18 Feb 2018

Review of "Mediterranean precipitation response to greenhouse gases and aerosols," by T. Tang et al., submitted to Atmospheric Chemistry and Physics.

This study intercompares model predictions of precipitation resulting from changes in black carbon, sulfate, and greenhouse gas forcings. While the issue is interesting, the problem with this study is that the paper virtually ignores any discussion of the intricate aerosol-cloud interactions that affect precipitation. Not only does the paper not even describe the aerosol-cloud interactions or the relevance of the mixing state and hygroscopicity of aerosols or of cloud microphysics, it is not clear to what extent any or all the models treat these processes. As such, it is impossible to determine whether the conclusions reached by the authors are reasonable because they don't even discuss if their models are appropriate for studying the issue.

[Figure]

Additional comments are given below.

Abstract. "The results from this study suggest that future BC emissions may significantly affect regional water resources, agricultural practices..." Whereas, this statement may or may not be correct, I don't agree that it is a conclusion of the present study because this study does not specify that it even considers the impacts of cloud activation of BC versus sulfate aerosol. The word activation is not even used in the paper.

Introduction. The authors are missing a major effect of dark aerosols, namely cloud absorption effects, which is the burning off of clouds due to absorption by black and brown carbon particles either within cloud drops or between them (Jacobson, 2012). The authors should mention this effect and discuss how it might affect results of the study if it were included, since it is one of the reasons clouds are thinner and precipitation is lower in highly polluted regions.

Introduction. "In addition to their influence on temperatures and precipitation, aerosols may also affect large-scale atmospheric circulation." The fundamental effect of aerosols on circulation starts with their reduction in near-surface wind speeds (Jacobson and Kaufmann, 2006).

Models. The impacts of aerosol particles on precipitation involve intricate and detailed physical processes, yet the paper treats such processes as a black-box subject. The information about the models provided in Table 1 is not sufficient to evaluate the models' ability to simulate the impacts of aerosol particles on precipitation. The information needed include the following parameters (ideally presented in a table), and it is not helpful to refer readers to other papers to dig out this information, particularly paper by paper.

1) How many aerosol modes or size bins? 2) How many aerosol components per mode or size bin, and what are the components? 3) Are aerosol particles treated as fully externally mixed, fully internally mixed, or evolving from externally to internally mixed. If

they evolve, do they evolve based on an empirical time constant or based on physical processes. 4) Which physical processes affect the aerosol size distribution? Homogeneous nuclear, coagulation, condensation, dissolution, reversible internal chemical reaction, dry deposition, sedimentation? 5) Do cloud drops physically activate on aerosol particles or is there an empirical relationship between the number of activated cloud drops and aerosol particles? 6) What is the assumed mixing state of black carbon for cloud activation purposes? Is it hygroscopic or hydrophobic? Are different sources of black carbon treated differently in terms of composition? 7) Are clouds treated as bulk parameters or are they treated with size modes or with size bins? 8) What physical processes affect cloud drop growth to precipitation particles? 9) Are clouds treated as subgrid phenomena in the GCM? How are they treated? How many clouds are allowed in each model grid column?

Once such information is provided, the authors should evaluate which models, if any, are most likely to provide reasonable results regarding the impacts of aerosol particles on precipitation.

Results. The authors provide end results of temperature change for a given emission or concentration but should discuss whether and how aerosol-cloud or cloud-cloud microphysical processes are treated and are affecting the results.

References Jacobson, M.Z., and Y.J. Kaufmann, Wind reduction by aerosol particles, Geophys. Res. Lett., 33, L24814, 2006

Jacobson, Investigating cloud absorption effects: Global absorption properties of black carbon, tar balls, and soil dust in clouds and aerosols, J. Geophys. Res., 117, D06205, 2012

―――――――――――――――――――

---

## Referee Comment (RC2) · Anonymous Referee #2 · 1 Mar 2018

This paper's focus is the local and large scale effects of radiative forcing by greenhouse gasses and aerosol on precipitation in the Mediterranean region. A decrease trend was observed for precipitation in this region during the last few decades and this study aims to explore the main processes behind this trend. To do so they use the outputs of set of climate models participating in the Precipitation Driver and Response Model Intercomparison Project.

This work suggests that both GHG and aerosols contribute to this decrease trend, by local as well as large scale effects. In particular, the contribution of shortwave absorption by black carbon (BC) is highlighted. Clear sky radiative effects are treated in details while hardly no attention is given to the aerosol effect on clouds' processes and properties (defined here in general as the indirect effect, although some of the models

do treat it). This work concludes that in addition to the local effects, BC absorption drives changes in large scale (global scale systems) such as enhanced positive North Atlantic Oscillation (NAO)/Arctic and it links it to a shift in the jet location (storm track) that implies drying of the Mediterranean and more rain over Europe. Let me start with the strength of the work. The insights on the link between the local BC absorption to the large-scale dynamics are interesting. Radiative effects on the local scale could be estimated by simpler radiation models but the derived effect on the dynamics could be resolved only by GCMs. Such dynamical results can be more important than the local effects and if all climate models show the same dynamical trend, it is important.

But even here, as in many (most) of the GCM studies, it is hard for someone who does not belong to the GCM community to evaluate this work. It is presented as model results and we have to believe it. One way to make such messages more approachable to all the climate researcher is to try to show the trend using as much as possible simpler models (toward an ideal GCM) such that the governing processes are demonstrated in a clearer way.

Apart from this, two main components are missing in this study: (1) The most important aerosol type over the Mediterranean is dust. Mostly Saharan dust. There are many studies that have shown how important are radiative and microphysical effect of Saharan dust. In this study which is dedicated to aerosol effects the word "dust" does not appear. Even if the authors want to focus on other processes they should first discuss dust in the introduction and explain why dust is not considered in this work. (2) On a similar note, since this paper deals with aerosols, clouds and precipitation, much more attention should be given to cloud aerosol interactions. Even if the authors estimate that this effect is negligible compared to other effects, they should invest efforts in proving it. They write in the conclusions part that the indirect and semi-direct effects are estimated to be small. I fail to understand how they know it and why they are so sure about it. I expect clouds to be extremely sensitive both to changes in the aerosol loading internally and to changes in the temperature (and RH) profiles due to

BC warming. Since rain is the sink of clouds it is not clear why such effects are less important.

---

## Author Comment (AC1) · 1 May 2018

We thank the anonymous reviewer for his/her helpful comments and suggestions. Please see our responses below:

This study intercompares model predictions of precipitation resulting from changes in black carbon, sulfate, and greenhouse gas forcings. While the issue is interesting, the problem with this study is that the paper virtually ignores any discussion of the intricate aerosol-cloud interactions that affect precipitation. Not only does the paper not even describe the aerosol-cloud interactions or the relevance of the mixing state and hygroscopicity of aerosols or of cloud microphysics, it is not clear to what extent any or all the models treat these processes. As such, it is impossible to determine whether the conclusions reached by the authors are reasonable because they don't even discuss if their models are appropriate for studying the issue.

Response: the energy budget analyses in our study show that large-scale dynamical responses rather than local responses (aerosol-cloud interactions) appear to dominate the precipitation change in the Mediterranean region. As a result, and due to the large spread in how the available models treat aerosol-cloud interactions, we did not emphasize aerosol-cloud interactions in our current study. However, how the GCMs treat the aerosols will be included in a table in the revised manuscript. The PDRMIP models used in the study are essentially the same as or similar to those used in the Coupled Model Inter-comparison Project Phase 5 (CMIP5) archive from the 5th report of Inter-governmental Panel on Climate Change. In the PDRMIP project, most of the models were run using climatological aerosols as a way to examine the similarity in model responses when driven with the same aerosol concentrations rather than including differences in both concentrations and responses. This leads to less realism in the physics, particularly of aerosol-cloud interactions, and hence the study focuses on aspects of the response that appear to be less sensitive to those interactions as they are relatively robust across the models despite some using interactive aerosols while others used climatological fields. This is now explicitly stated in the paper (section 2.1). Undoubtedly, analyses with the setup used in PDRMIP are not perfect, but still useful and these have now been widely accepted as such by the peer-review process. In the present case, the similarity between the results of models with detailed representation of aerosol-cloud interactions and those without such processes suggests that those may not play a major role in the precipitation response that we focus on in this work, though we agree that this topic is worth additional future study.

Additional comments are given below.
**Abstract**. "The results from this study suggest that future BC emissions may significantly affect regional water resources, agricultural practices…" Whereas, this statement may or may not be correct, I don't agree that it is a conclusion of the present study because this study does not specify that it even considers the impacts of cloud activation of BC versus sulfate aerosol. The word activation is not even used in the paper.

Response: this conclusion is based on the results in section 3 (Fig. 1), which indicates a higher sensitivity of precipitation response in the Mediterranean region to BC forcing. As a result, changes in BC concentrations could greatly impact precipitation and thus, water resources in this region. Aerosol-cloud interactions including cloud activation that takes into account differences between BC and $SO_4$ are included in most of the PDRMIP models in this study, and thus, in the results for section 3, although these are not represented in as much detail as in the most

sophisticated models (which are too expensive to be used in CMIP-type analyses). As noted above, the similarity in the results of models with relatively detailed representation of aerosol-cloud interactions and those without such processes suggests that those may not play a major role in the precipitation response that we focus on in this work.

**Introduction**. The authors are missing a major effect of dark aerosols, namely cloud absorption effects, which is the burning off of clouds due to absorption by black and brown carbon particles either within cloud drops or between them (Jacobson, 2012). The authors should mention this effect and discuss how it might affect results of the study if it were included, since it is one of the reasons clouds are thinner and precipitation is lower in highly polluted regions.
Response: Admittedly, these effects are not included in the PDRMIP models nor any other current CMIP-class GCMs, which may impact the results. As noted in the discussion section, those effects generally have a small overall forcing with a large uncertainty range. This limitation, including the reference, has been included in the discussion section in the revised version.

**Introduction**. "In addition to their influence on temperatures and precipitation, aerosols may also affect large-scale atmospheric circulation." The fundamental effect of aerosols on circulation starts with their reduction in near-surface wind speeds (Jacobson and Kaufmann, 2006).
Response: thanks for the reference. It is included in the revised manuscript.

**Models**. The impacts of aerosol particles on precipitation involve intricate and detailed physical processes, yet the paper treats such processes as a black-box subject. The information about the models provided in Table 1 is not sufficient to evaluate the models' ability to simulate the impacts of aerosol particles on precipitation. The information needed include the following parameters (ideally presented in a table), and it is not helpful to refer readers to other papers to dig out this information, particularly paper by paper.

1) How many aerosol modes or size bins?
 Please see the table below.

2) How many aerosol components per mode or size bin, and what are the components?
Please see the table below.

3) Are aerosol particles treated as fully externally mixed, fully internally mixed, or evolving from externally to internally mixed. If they evolve, do they evolve based on an empirical time constant or based on physical processes.
Please see the table below.

4) Which physical processes affect the aerosol size distribution? Homogeneous nuclear, coagulation, condensation, dissolution, reversible internal chemical reaction, dry deposition, sedimentation?
Response: all these processes affect the aerosol size distribution, which varies model by model in terms of which aerosols are represented as a function of size (see the table).

5) Do cloud drops physically activate on aerosol particles or is there an empirical relationship between the number of activated cloud drops and aerosol particles?
Please see the table below.

6) What is the assumed mixing state of black carbon for cloud activation purposes? Is it hygroscopic or hydrophobic? Are different sources of black carbon treated differently in terms of composition?
Please see the table below.

| Model | Size bin/aerosol mode | Aerosol per size bin | Mixing state | Evolve Empirically or physically | Aerosol size distribution | Cloud drop activate physically or empirically | interactive vs climatological aerosols |
|---|---|---|---|---|---|---|---|
| CanESM | S, N, BC, dust, SS, OC | N/A | Internal | N/A | Log-normal | N/A | Interactive |
| GISS | S (1), N (1), OC (1), BC (1), SS (2), dust (4) | N/A | Internal & external | N/A | Log-normal | Empirically | Climatological |
| HadGEM2 | S: 3 modes (Aitken, accumulation, dissolved) BC, OC, BB: 3 modes (fresh, aged, dissolved/in-cloud) SS: 2 modes (jet, film) Dust: 6 size bins | N/A | External | Physically | Log-normal | Empirically | Interactive |
| HadGEM3 | S: 3 modes (Aitken, accumulation, dissolved) BC, OC, BB: 3 modes (fresh, aged, dissolved/in-cloud) SS: 2 modes (jet, film) Dust: 6 size | N/A | External | N/A | Prescribed log-normal distribution for radiation | Empirically | Climatological |

| | | | | | | | |
|---|---|---|---|---|---|---|---|
| | bins | | | | | | |
| MIROC | S (1), BC (1), OC (1), dust (6), SS (4) | N/A | Internal & external | N/A | Prescribed log-normal for radiation and diagnosing number concentration | Based on the Köhler theory (Abdul-Razzak and Ghan 2000) | Interactive |
| CAM4 | S, SS (4 size bins), dust (4 size bins), BC (2 modes), POM (2 modes), SOA | Fixed sizes | External | N/A | Log-normal | N/A | Climatological |
| CAM5 | S, POM, SOA, SS, BC, dust (3 modes) | Aitken: S, SOM, SS Accumulation: S, POM, SOM, BC, dust, SS Coarse: dust, SS, S | Internal | Physically | Log-normal | Physically | Interactive |
| NorESM | 13modes, 44 size bins, S, OM, BC, SS, dust | | Internal & external | Physically | Log-normal | Physically | Interactive |
| IPSL | S, BC, OC, dust, SS | N/A | External | N/A | Prescribed log-normal for radiation | Empirically | Climatological |

S = sulfate, N = nitrate; SS= sea salt, OC = organic carbon, BC = black carbon, OM = organic mass, BB = biomass burning, SOA = secondary organic aerosol, POM = primary organic matter

7) Are clouds treated as bulk parameters or are they treated with size modes or with size bins?
Response: clouds are generally bulk parameterized in GCMs.

8) What physical processes affect cloud drop growth to precipitation particles?
Response: condensation, evaporation and coalescence.

9) Are clouds treated as subgrid phenomena in the GCM? How are they treated? How many clouds are allowed in each model grid column?
Response: clouds are treated as subgrid in GCMs. Cloud amount (fraction) is generally parameterized on the basis of meteorological conditions, such as relative humidity, atmospheric stability and convections. Cloud amount in each grid varies by model.

Once such information is provided, the authors should evaluate which models, if any, are most likely to provide reasonable results regarding the impacts of aerosol particles on precipitation.
Response: the emission-driven models are potentially more realistic in this regard. However, the results are similar across the models and no significant differences are observed between the models using interactive and those using climatological aerosols. Thus, all models are included, and indeed this underlies our conclusion that the results are generally associated with large-scale dynamics are less sensitive to details of local aerosol-cloud interactions.

**Results**. The authors provide end results of temperature change for a given emission or concentration but should discuss whether and how aerosol-cloud or cloud-cloud microphysical processes are treated and are affecting the results.
Response: the temperature change is out of the scope of this paper. We didn't include any results for temperature change. For aerosol-cloud processes, we have added related information (table) and discussion in the revised manuscript.

References Jacobson, M.Z., and Y.J. Kaufmann, Wind reduction by aerosol particles, Geophys. Res. Lett., 33, L24814, 2006

Jacobson, Investigating cloud absorption effects: Global absorption properties of black carbon, tar balls, and soil dust in clouds and aerosols, J. Geophys. Res., 117, D06205, 2012

---

## Author Comment (AC2) · 1 May 2018

We thank the anonymous reviewer for his/her helpful comments and suggestions. Please see our responses below.

This paper's focus is the local and large-scale effects of radiative forcing by greenhouse gasses and aerosol on precipitation in the Mediterranean region. A decrease trend was observed for precipitation in this region during the last few decades and this study aims to explore the main processes behind this trend. To do so they use the outputs of set of climate models participating in the Precipitation Driver and Response Model Intercomparison Project.

This work suggests that both GHG and aerosols contribute to this decrease trend, by local as well as large scale effects. In particular, the contribution of shortwave absorption by black carbon (BC) is highlighted. Clear sky radiative effects are treated in details while hardly no attention is given to the aerosol effect on clouds' processes and properties (defined here in general as the indirect effect, although some of the models do treat it). This work concludes that in addition to the local effects, BC absorption drives changes in large scale (global scale systems) such as enhanced positive North Atlantic Oscillation (NAO)/Arctic and it links it to a shift in the jet location (storm track) that implies drying of the Mediterranean and more rain over Europe. Let me start with the strength of the work. The insights on the link between the local BC absorption to the large-scale dynamics are interesting. Radiative effects on the local scale could be estimated by simpler radiation models but the derived effect on the dynamics could be resolved only by GCMs. Such dynamical results can be more important than the local effects and if all climate models show the same dynamical trend, it is important.
Response: thanks.

But even here, as in many (most) of the GCM studies, it is hard for someone who does not belong to the GCM community to evaluate this work. It is presented as model results and we have to believe it. One way to make such messages more approachable to all the climate researcher is to try to show the trend using as much as possible simpler models (toward an ideal GCM) such that the governing processes are demonstrated in a clearer way.
Response: the climate system is extremely complex and constitutes of many components. The PDRMIP models used in this study are the latest state-of-the-art GCMs, and they are the same or similar versions to those used in the latest IPCC report (Flato et al., 2013), representing the best global climate models in the community. Here we argue that models with finer resolutions and more components generally produce more realistic results than simple ideal models. We agree, however, that there is value in using simpler models such as those developed as companions alongside the new generation of the NCAR GCM. Such models, however, were not available for the versions used in PDRMIP and no groups using simpler models participated.

Apart from this, two main components are missing in this study:
(1) The most important aerosol type over the Mediterranean is dust. Mostly Saharan dust. There are many studies that have shown how important are radiative and microphysical effect of Saharan dust. In this study which is dedicated to aerosol effects the word "dust" does not appear. Even if the authors want to focus on other processes they should first discuss dust in the introduction and explain why dust is not considered in this work.

Response: dust aerosols are included in the model, but not in the experiments. We didn't include dust in the PDRMIP study, since we are only investigating the impacts of anthropogenic forcing. We added a comment on this in the revised manuscript.

(2) On a similar note, since this paper deals with aerosols, clouds and precipitation, much more attention should be given to cloud aerosol interactions. Even if the authors estimate that this effect is negligible compared to other effects, they should invest efforts in proving it. They write in the conclusions part that the indirect and semi-direct effects are estimated to be small. I fail to understand how they know it and why they are so sure about it. I expect clouds to be extremely sensitive both to changes in the aerosol loading internally and to changes in the temperature (and RH) profiles due to BC warming. Since rain is the sink of clouds it is not clear why such effects are less important.

Response: the energy budget analysis in the original manuscript (Fig. 3 and Fig. 6) shows that the local effects such as clouds, radiation (absorption and scattering) are overwhelmed by large-scale dynamical responses in the Mediterranean region. As a result, we place more emphasis on the dynamical responses instead of local responses. However, these large-scale changes are the result of remote forcings, the magnitudes of which may still depend on the inclusion of semi-direct and indirect effects. So even if local effects are not that important, it doesn't necessarily mean that indirect effects are not important in driving the responses. In fact, semidirect and indirect effects are included in most of the PDRMIP models, and thus the results in section 3. However, in the PDRMIP project, many of the models were run using climatological aerosols as a way to examine the similarity in model responses when driven with the same aerosol concentrations rather than including differences in both concentrations and responses. This leads to less realism in the physics, particularly of aerosol-cloud interactions, and hence the study focuses on aspects of the response that appear to be less sensitive to those interactions as they are relatively robust across the models despite some using interactive aerosols while others used climatological fields. This is now explicitly stated in the paper (section 2.1). In particular, in the present case in which we examine responses in the Mediterranean, the similarity between the models with detailed representation of aerosol-cloud interactions and those without give similar results (e.g. Figure 2 in the paper, panels b and c where there is no clear difference between climatological and interactive models), suggesting that such processes may not play a major role in this precipitation response. In addition, indirect effects are not complete in any of the models. For example, ice particles and internal cloud absorption are not included. For BC, these effects contribute to an overall small forcing with a very large uncertainty, so these are discussed but cannot be explicitly included for BC. For $SO_4$, the forcing could be as much as doubled when accounting for aerosol-cloud interactions, and thus the impact on precipitation could also be doubled if including them, but this would still be a small impact, as noted in the discussion section.

**References**

Flato, G., J. Marotzke, B. Abiodun, P. Braconnot, S.C. Chou, W. Collins, P. Cox, F. Driouech, S. Emori, V. Eyring, C. Forest, P. Gleckler, E. Guilyardi, C. Jakob, V. Kattsov, C. Reason and M. Rummukainen, 2013: Evaluation of Climate Models. In: Climate Change 2013: The Physical Science Basis. Contribution of Working Group I to the Fifth Assessment Report of the Intergovernmental Panel on Climate Change [Stocker, T.F., D. Qin, G.-K. Plattner, M. Tignor, S.K. Allen, J. Boschung, A. Nauels, Y. Xia, V. Bex and P.M. Midgley (eds.)]. Cambridge University Press, Cambridge, United Kingdom and New York, NY, USA.

---

## Author Response (AR2)

**Suggestions for revision or reasons for rejection (will be published if the paper is accepted for final publication)**

Re-review of "Mediterranean precipitation response to greenhouse gases and aerosols," by T. Tang et al., submitted to Atmospheric Chemistry and Physics.

The authors have responded to the referees, but have hardly addressed in the text the main issue, pointed out by both referees, that the models used do not treat aerosol-cloud interactions in any detail, so the results could change significantly if they did.

To remedy this problem, the authors need to (1) reflect more accurately in the text and the title of their manuscript that their study looks at the dynamical rather than microphysical response of aerosols and (2) be more transparent in the text about what aerosol-cloud treatments are and are not included in the models.

First, the title and abstract need to be changed to reflect accurately what the simulation performed is doing. Since aerosol-cloud interactions are either hardly treated or treated weakly in all models, as acknowledge by the authors, the study is really about the dynamical meteorological responses, not the cloud microphysical responses, of different aerosol loadings to precipitation.
Response: accepted. Title and abstract revised and more description included in section 2.1 to clarify that the modeling was not designed to study aerosol-cloud interactions but to look at dynamic responses.

Thus, the title should be changed to "Climate model dynamical response to greenhouse gases and aerosols in the Mediterranean" so as not to confuse readers into thinking the study treats microphysical responses, which it does not to any realistic degree.
Response: accepted, changed along lines suggested.

Similarly, the abstract should be changed to "Here, we compare the modeled dynamical response to individual forcing agents in the Mediterranean using a set of global climate models."
Response: accepted.

Please do not state that the models used are "State of the art," when none of the models used treat aerosol-cloud interactions in nearly the detail as some other models.
Response: deleted.

Please explain clearly in the abstract that the models do not include detailed aerosol-cloud microphysical interactions and state specifically that inclusion of such processes may change the results.
Response: included statements on both these issues.

Please state in the manuscript itself which aerosol-cloud processes are missing.
Response: included in the section 2.1 and discussion sections.

The PDRMIP project aims to examine the similarity in model responses when driven with the same aerosol concentrations rather than including differences in both concentrations and responses. The aerosol-cloud interactions are not fully represented in the majority of models as these are concentration-driven. In some

concentration-driven models the aerosol-cloud interactions are switched off, but even if they are on the aerosol concentrations are fixed so that these interactions are not realistic. These processes are included in some of the emissions-driven models, but these are a minority of the project.
In other words, aerosol-cloud interactions are not in the scope of the PDRMIP project though there are other projects investigating those (e.g. AeroCom). Hence this study focuses on aspects of the response that appear to be less sensitive to those interactions given that they are relatively robust across the models (despite some using interactive aerosols while others used climatological fields).

Please add the table from the response to referees describing the aerosol/cloud activation properties of each model in the manuscript itself.
Response: included.

If these steps are taken, at least it will be clear to readers that the responses examined are not the complete responses and where further work is needed.
Response: agreed. Thanks for ensuring that this is clear in the description of the modeling.

**Suggestions for revision or reasons for rejection (will be published if the paper is accepted for final publication)**

It would be nice to have a more process level understanding of the trends shown here.
Response: the PDRMIP project aims to examine the similarity in model responses when driven with the same aerosol concentrations rather than including differences in both concentrations and responses. The aerosol-cloud interactions are turned off in some concentration-driven models. For the concentration-driven models with aerosol-cloud interactions switched on, the fact that the aerosol concentrations are fixed means they cannot fully interact with clouds (e.g. even if they get rained out, they persist).
In other words, process level understanding of aerosol-cloud interactions is not within the scope of PDRMIP project and there are already other projects investigating aerosol-cloud interactions (e.g. AeroCom). Hence this study focuses on dynamic aspects of the response that appear to be less sensitive to those interactions as they are relatively robust across the models (despite some using interactive aerosols while others used climatological fields), and hence we present analyses of both energy budgets and dynamic responses in these models.

The same for dust and cloud-aerosol interactions. Note that some changes in dust loading and patterns can be attributed to anthropogenic changes.
Response: dust is included in the background aerosols in the PDRMIP experiment. However, anthropogenic changes in dust loadings (e.g., land use/cover change) are not part of the PDRMIP simulations as the overarching goal of the MIP was to examine the response to prescribed changes in GHGs, sulfate and BC (not dust) and not to include interactive aerosol responses to climate variability or change. Hence there are no PDRMIP results relevant to the question of the role of changes in dust on Mediterranean precipitation. We've added a sentence to the text stating that it could be useful for future work to examine this.

Even if the collective GCM runs suggest that cloud-aerosol interactions and aerosol radiative effects are minor, I recommend that the authors will provide more details on them in the paper.
Response: Although the cloud-aerosol interactions appear to be minor in these results, the aerosol radiative effects are important and are already documented in the paper. Without them, there would be no impacts from aerosols. Aerosol radiative effects are discussed in the energy budget analysis and dynamical responses, and the revised paper points out explicitly that aerosol-cloud interactions do not seem to be important but that many of the models did not fully represent these and so this is another area meriting future work.

**List of changes**

1) Changes in the title, abstract (review #1)

2) More descriptions about the aerosol treatment, mainly in section 2.1 and Table 1 (review #1 and 2)

3) The reasons of excluding Saharan dust are included in the section 2.1 (review #2)

[revised manuscript text omitted]